# Dietary Supplementation throughout Life with Non-Digestible Oligosaccharides and/or n-3 Poly-Unsaturated Fatty Acids in Healthy Mice Modulates the Gut–Immune System–Brain Axis

**DOI:** 10.3390/nu14010173

**Published:** 2021-12-30

**Authors:** Kirsten Szklany, Phillip A. Engen, Ankur Naqib, Stefan J. Green, Ali Keshavarzian, Alejandro Lopez Rincon, Cynthia J. Siebrand, Mara A. P. Diks, Melanie van de Kaa, Johan Garssen, Leon M. J. Knippels, Aletta D. Kraneveld

**Affiliations:** 1Division of Pharmacology, Utrecht Institute of Pharmaceutical Sciences, Faculty of Science, Utrecht University, 3584 CG Utrecht, The Netherlands; k.szklany@uu.nl (K.S.); a.lopezrincon@uu.nl (A.L.R.); cjsieb@hotmail.com (C.J.S.); m.a.p.diks@uu.nl (M.A.P.D.); mel_kaa@live.nl (M.v.d.K.); J.garssen@uu.nl (J.G.); leon.knippels@danone.com (L.M.J.K.); 2Rush Center for Integrated Microbiome and Chronobiology Research, Rush Medical College, Rush University Medical Center, Chicago, IL 60602, USA; Phillip_Engen@rush.edu (P.A.E.); Ankur_Naqib@rush.edu (A.N.); Ali_Keshavarzian@rush.edu (A.K.); 3Genomics and Microbiome Core Facility, Rush University Medical Center, Chicago, IL 60602, USA; Stefan_Green@rush.edu; 4Department of Medicine & Physiology, Rush University Medical Center, Chicago, IL 60602, USA; 5Department of Data Science, Julius Center for Health Sciences and Primary Care, University Medical Center Utrecht, 3584 EA Utrecht, The Netherlands; 6Global Centre of Excellence Immunology, Nutricia Danone Research, 3584 CT Utrecht, The Netherlands

**Keywords:** behaviour, galacto-oligosaccharide, fructo-oligosaccharide, prebiotics, omega-3 fatty acids, PUFA, intestinal microbiota, SCFA, healthy mice, early life

## Abstract

The composition and activity of the intestinal microbial community structures can be beneficially modulated by nutritional components such as non-digestible oligosaccharides and omega-3 poly-unsaturated fatty acids (n-3 PUFAs). These components affect immune function, brain development and behaviour. We investigated the additive effect of a dietary combination of scGOS:lcFOS and n-3 PUFAs on caecal content microbial community structures and development of the immune system, brain and behaviour from day of birth to early adulthood in healthy mice. Male BALB/cByJ mice received a control or enriched diet with a combination of scGOS:lcFOS (9:1) and 6% tuna oil (n-3 PUFAs) or individually scGOS:lcFOS (9:1) or 6% tuna oil (n-3 PUFAs). Behaviour, caecal content microbiota composition, short-chain fatty acid levels, brain monoamine levels, enterochromaffin cells and immune parameters in the mesenteric lymph nodes (MLN) and spleen were assessed. Caecal content microbial community structures displayed differences between the control and dietary groups, and between the dietary groups. Compared to control diet, the scGOS:lcFOS and combination diets increased caecal saccharolytic fermentation activity. The diets enhanced the number of enterochromaffin cells. The combination diet had no effects on the immune cells. Although the dietary effect on behaviour was limited, serotonin and serotonin metabolite levels in the amygdala were increased in the combination diet group. The combination and individual interventions affected caecal content microbial profiles, but had limited effects on behaviour and the immune system. No apparent additive effect was observed when scGOS:lcFOS and n-3 PUFAs were combined. The results suggest that scGOS:lcFOS and n-3 PUFAs together create a balance—the best of both in a healthy host.

## 1. Introduction

The gut microbiota is the total collection of microbial organisms within a community. The microbiota evolves and adapts to its host over a lifetime and microbiota activities have significant consequences for the host in terms of health and disease. The development of the intestinal microbiota during early life appears to be essential in the maturation of the immune system, adaptation of intestinal tissue morphology, as well as the development of the brain and behaviour [1,2,3]. In the first phase of life, microbiota development can be modified by several factors, such as mode of delivery (caesarean section versus vaginal delivery), antibiotic use and mode of feeding (formula versus breastfeeding) [4]. Consequently, impaired early-life development of the gut microbiota could lead to a disturbed maturation of the immune system and brain, possibly resulting in increased risk of developing immune and brain disorders, respectively [1]. Therefore, healthy development of the intestinal microbiota is important.

For the maturation of the immune system, exposure to environmental factors such as food antigens is essential. This environmental exposure is necessary for the immune system to develop tolerance towards harmless components. The precise mechanism of tolerance development is unknown, but the differentiation of regulatory T cells (Tregs) in Peyer’s patches and mesenteric lymph nodes (MLNs) could play an important role [5]. As mentioned above, the maturation of the immune system is also influenced by the intestinal microbiota. Short-chain fatty acids (SCFAs) play a role in the development of mucosal tolerance of food antigens. The SCFAs are metabolites from the intestinal microbiota fermentation of carbohydrates. They have immunomodulatory capacities and are able to induce the differentiation and activity of Tregs, which in turn can result in tolerance development [6,7]. In the absence of an intestinal commensal microbiota, the immune system is deprived. This deprivation is manifested by functional defect Tregs and an exaggerated systemic type-2 immune response [8,9]. These data indicate the essential role of the intestinal microbiota in immune system development.

It is widely accepted that the intestinal microbiota plays an important role in the development of the brain and behaviour. Animals without intestinal microbes, such as germ-free mice and antibiotic-treated animals, have shown reduced anxiety-like behaviour and impaired social behaviour [10,11,12]. These altered behaviours in germ-free mice are accompanied by disturbed neurotransmission in the central nervous system, where serotonin levels are increased compared to conventional mice [12,13]. To emphasise the importance of the intestinal microbiota in behaviour, these behavioural deficits were alleviated after microbial re-colonisation [10,11,12]. To imply causality of the modulation of behaviour by intestinal microbiota in neurobehavioural disorders, faecal microbiota transplant (FMT) has been used. FMT from attention-deficit hyperactivity disorder and autism spectrum disorder (ASD) individuals into germ-free mice resulted in increased anxiety, repetitive and impaired social behaviour [14,15]. Transfer of anxiety and depression behavioural traits also occurred via the intestinal microbiota in both mice and rats [16]. Moreover, in preclinical models of autism spectrum, depression and anxiety disorders, administration of specific bacteria restored the behavioural deficits [16].

Tryptophan is an essential amino acid that is metabolised into indole, serotonin (5-HT) and kynurenine. In the intestine, tryptophan first encounters bacteria, some of these bacteria are tryptophan utilising and metabolise tryptophan into 5-HT and/or indole [17,18]. Indole is able to modulate host physiology, for example integrity of intestinal epithelial barrier [19]. Then, tryptophan encounters intestinal epithelial cells, where tryptophan in enterochromaffin cells is converted into 5-HT. The basolateral released 5-HT interacts with the enteric nervous system and, through vagal afferent nerve signalling, reaches the central nervous system [17]. The secretion of 5-HT from enterochromaffin cells can be mediated by SCFA and thus indirectly by intestinal bacteria [20]. Lastly, most of the tryptophan available is metabolised through the kynurenine pathway. Kynurenine is further metabolised into neuroprotective kynurenic acid and neurotoxic quinolinic acid [18]. The intestinal microbiota might indirectly affect the kynurenine pathway [19]; for example, *L. reuteri* was associated with decreased circulating kynurenine and normalised stress-induced behaviour [21]. In germ-free mice, the levels of plasma 5-HT and tryptophan are elevated, indicating the reduced metabolism of tryptophan [18,19].

It is well acknowledged that diet plays a significant role in shaping the intestinal microbiota. Microbial community structures can be beneficially modulated by nutritional components such as non-digestible oligosaccharides (NDOs) with prebiotic function and omega-3 poly-unsaturated fatty acids (n-3 PUFAs) [22,23,24,25,26]. These dietary components have been shown to be important in immune function as well as brain development and behaviour [27,28,29,30,31].

Specific NDOs in human breast milk, called human milk oligosaccharides (HMOs), are the third-most abundant milk solid component in human breast milk. These prebiotics are essential in the development of the immune system [32]. In infant formula, which is the alternative to breastfeeding, NDOs such as a mixture of short-chain galacto-oligosaccharide (scGOS) and long-chain fructo-oligosaccharide (lcFOS) are added to mimic the modulatory effects of HMOs. scGOS:lcFOS modulates the activity and growth of intestinal bacteria to shape a healthy intestinal microbiota [23,26,33]. Additionally, scGOS:lcFOS modulates the immune system directly, i.e., by upregulating IL-10 generation by DCs and inducing Tregs [34,35]. This is important to stimulate development of tolerance and for skewing from a Th2 response at birth towards a Th1 response [36,37]. This skewing might decrease the risk of developing chronic inflammatory diseases such as allergies. Prebiotics also have the capacity to influence brain development and behaviour [31,38,39,40]. Specific prebiotics are indicated to affect the serotonergic system and to modulate behaviour in in vivo models, where social behaviour is improved and anxiety-like and depression-like behaviour are reduced [31,38,40].

n-3 PUFAs play an important role in the development of the immune system and the brain and, according to ISAPP, are prebiotic candidates [41]. These fatty acids are known to exert anti-inflammatory effects and also induce the skewing from a Th2 to a Th1 immune response early in life, which, as mentioned above, is important to reduce the risk of developing allergies [42,43,44]. n-3 PUFAs are essential in the development of the brain. They are incorporated in the neuronal cell membrane and play an essential role in supporting brain function throughout life [45,46]. Two of the most important n-3 PUFAs are eicosapentaenoic acid (EPA) and docosahexaenoic acid (DHA). Deficiencies in n-3 PUFAs in mice have resulted in depression-like behaviour and impaired social behaviour [25,47]. Moreover, both EPA and DHA supplementation in rats have led to reduced anxiety-like behaviour [48]. Although there is limited evidence, n-3 PUFAs are probably able to modulate intestinal microbiota composition. Dietary supplementation with n-3 PUFAs resulted in changed faecal bacterial taxa, depicting lower abundances of the genus *Coprococcus* and higher abundances of the genera *Bifidobacterium*, *Oscillospira* and *Lactobacillus* [25,49]. These data might indicate that, in addition to affecting the immune system and brain development directly, n-3 fatty acids are able to influence these systems through microbial modulation of the intestinal microbiota [25,49].

Based on the data that both scGOS:lcFOS and n-3 PUFAs modulate the intestinal microbiota and beneficially affect both the immune system and brain development, we hypothesised that a combination of scGOS:lcFOS and n-3 PUFAs can result in an additive effect. To evaluate this hypothesis, we investigated the effect of a combined dietary mixture of scGOS:lcFOS and n-3 PUFAs on caecal content microbiota and activity and the development of the immune system, the brain and behaviour from day of birth in healthy mice.

## 2. Materials and Methods

### 2.1. Animals

Sixteen-day pregnant BALB/cByJ were purchased from Charles River Laboratories (Maastricht, The Netherlands) and housed individually. The dams were allocated to the control (*n* = 3, male pups *n* = 8), scGOS:lcFOS (*n* = 4, male pups *n* = 10), *n*-3 PUFAs (*n* = 3, male pups *n* = 11) or combination (*n* = 4, male pups *n* = 15) diet groups on the day their pups were born (postnatal day 0). The male mice were weaned on postnatal day 21 and continued the same diet as their mother until the end of the experiment (control *n* = 8, the supplementary diet groups *n* = 10, *n* = 1–5 from each litter). Fourteen age-matched male BALB/cByJ were purchased from Charles River to be used as interaction mice in the social interaction behavioural test (see behavioural tests). The male mice were housed in groups after weaning. All mice were housed in Makrolon IIL cages and had ad libitum access to food and water. A light/dark cycle of 12 h was followed and the experimental procedures, including the behavioural tests, were performed during the light phase. All animal experimental procedures were carried out in compliance with national legislation following the EU Directive for the protection of animals used for scientific purposes and were approved by the Ethical Committee for Animal Research of Utrecht University (Approval number DEC 2014.I.12.090).

### 2.2. The Diets

The enriched diets were based on AIN-93G diets with a fat percentage of 10%. The scGOS:lcFOS diet consisted of 3% scGOS (degree of polymerisation 2–8, Friesland Campina, The Netherlands) and lcFOS (degree of polymerisation on average ≥23, Orafti, Wijchen, The Netherlands) in a 9:1 (*w*/*w*) ratio. In the n-3 PUFA diet, 6% of the soybean oil was substituted with 6% tuna oil. The composition of the experimental diets is shown in Appendix A. The tuna oil was a kind gift from Bioriginal (Den Bommel, The Netherlands). The combination diet contained 3% of the scGOS:lcFOS mixture and 6% tuna oil. The supplementations were added in an isocaloric manner. The diets were obtained from Ssniff Spezialdiäten gmbH (Soest, Germany).

### 2.3. Experimental Design

A schematic overview of the experimental design is illustrated in Figure 1. Behavioural tests for social, explorative, stereotypic and anxiety-like behaviour were conducted during adolescence (6 weeks old) and early adulthood (8 weeks old). After decapitation, the brain, intestines, caecum content, MLNs and spleen were isolated for further analysis. For MLN isolation, the abdominal cavity was opened, the caecum was lifted, and the superior mesenteric lymph nodes were isolated. Working towards the middle of the mesentery, the inferior lymph nodes were isolated.

### 2.4. Microbiota Profiling and Bioinformatics Analyses

Total DNA was extracted from mice caecal contents utilising the FastDNA bead-beating Spin Kit for Soil (MP Biomedicals, Solon, OH, USA), and DNA concentrations were measured via fluorometric quantitation (Qubit 1.0, Life Technologies, Grand Island, NY, USA). Primers 515F (Caporaso)–806R (Caporaso) targeting the variable region four (V4) of microbial small subunit (SSU or 16S) ribosomal RNA (rRNA) genes were used for PCR [50], and prepared for high-throughput amplicon sequencing using a modified two-step targeted amplicon sequencing (TAS) approach [51]. Negative controls were used with each set of amplifications, which indicated no contamination. Samples were pooled in equal volume using an EpMotion5075 liquid handling robot (Eppendorf, Hamburg, Germany). The library pool was purified using an AMPure XP cleanup protocol (0.6×, *v*/*v*; Agencourt, Beckmann-Coulter) to remove fragments smaller than 300 bp. The pooled libraries, with a 20% phiX spike-in, were loaded onto an Illumina MiniSeq (Illumina, San Diego, CA, USA) mid-output flow cell (2 × 153 paired-end reads) and sequenced using Fluidigm sequencing primers. Based on the distribution of reads per barcode, the amplicons (before purification) were re-pooled to generate a more balanced distribution of reads. The re-pooled and re-purified libraries were then sequenced on a high-output MiniSeq run (2 × 153 paired-end reads). Library preparation, pooling, and sequencing were performed at the Genome Research Core (GRC) at the University of Illinois at Chicago (UIC). Raw sequence data (FASTQ files) were deposited in the National Center for Biotechnology Information (NCBI) Sequence Read Archive (SRA), under the BioProject identifier PRJNA701436.

Raw FASTQ files for each sample were merged using the software package PEAR (Paired-end-read merger) (v0.9.8) (Dalhousie University, Halifax, Nova Scotia, NS, Canada) (http://www.exelixis-lab.org/web/software/pear (accessed on 28 December 2021)) [52,53]. Merged reads were quality trimmed, and primer sequences removed. Sequences shorter than 250 bases were discarded (CLC Genomics Workbench, v10.0, CLC Bio, Qiagen, Boston, MA, USA). Sequences were screened for chimeras (usearch8.1 algorithm) [54], and putative chimeric sequences were removed from the dataset (QIIME v1.8, Quantitative Insights Into Microbial Ecology, Knight Lab at the University of Colorado at Boulder, Boulder, CO, USA) [55]. Each sample was rarefied (45,000 sequences/sample) and data were pooled, renamed, and clustered into operational taxonomic units (OTU) at 97% similarity (usearch8.1 algorithm). Representative sequences from each OTU were extracted and classified using the uclust consensus taxonomy assigner (Greengenes 13_8 reference database). A biological observation matrix (BIOM) [56] was generated at each taxonomic level from phylum to species (“make OTU table” algorithm) and analysed and visualised using the software packages Primer7 [57] (PRIMER-E Ltd., Lutton, UK) and the R programming environment [58].

### 2.5. Caecum Short-Chain Fatty Acid Levels

The levels of SCFAs were analysed as previously described [59]. In short, the caecal contents were stored at −80 °C until analysis. After being defrosted on ice, the samples were diluted in PBS and homogenised followed by centrifugation for 10 min at 14,000× *g*. Next, the supernatant was heated for 10 min at 100 °C to inactivate all enzymes and centrifuged again. The SCFAs acetic, propionic, butyric, iso-butyric, valeric and iso-valeric acids were quantitatively determined by gas chromatography using a Shimadzu GC2010 gas chromatograph (Shimadzu Corporation, Kyoto, Japan) as described previously [59].

### 2.6. Immunohistochemistry Analysis of 5-HT-Positive Cells in Jejunum, Ileum and Colon

The intestinal 5-HT-positive cells were determined following the protocol previously described [60]. After isolation jejunal, ileal and colonic tissue (*n* = 6 of each tissue) were opened longitudinally, rolled in the direction from distal to proximal and fixed in 10% formalin for at least 24 h and embedded in paraffin. The 5 µm tissue sections (8 tissue sections from each swiss roll) were incubated in 0.3% H_2_O_2_ in methanol for 30 min to block endogenous peroxidase activity. Sections were rehydrated in ethanol and incubated with Proteinase K (DAKO, Code S3020). Non-specific staining was blocked with 5% goat serum, and sections were incubated overnight at 4 °C with the primary antibody mouse anti-5-HT (DAKO, Code M0758) 1:100. The following day, sections were incubated with biotinylated goat anti-mouse (DAKO, Code E0433) 1:200, followed by incubation with sABC complex 1:100. Staining was visualised by incubating the sections in the dark with 1× DAB solution for 10 min at room temperature. Nuclear staining with Mayer’s haematoxylin has been performed for 10 s. Digital images were captured using the software Image Pro (Media Cybernetics, Rockville, MD, USA) and an Olympus BX50 light microscope with a Leica DFC 320 digital camera. In colonic tissue, sections in the epithelial layer were counted in 10 consecutive crypts on five different places per tissue section. These colonic crypts covered 461 × 187 µm (8.6 × 104 µm^2^). For tissue sections of the jejunum and ileum, 5-HT-positive cells were counted in the same area size of 461 × 187 µm (8.6 × 104 µm^2^) on five different places per section. Due to differences in crypt sizes, using a consistent area size was the most practical way to make comparable measurements.

### 2.7. Cell Isolation from MLNs and Spleen

Single-cell suspensions of the MLNs and spleen were obtained by crushing the organs through a 70 µm cell strainer. The splenocytes were incubated with a lysis buffer (8.3 g/L NH_4_Cl, 1 g/L KHCO_3_, and 37.2 mg/L EDTA) to lyse the red blood cells. The cell suspensions were resuspended in PBS + 1% BSA (Sigma-Aldrich, St. Louis, MO, USA).

### 2.8. Flow Cytometry Analysis of the Immune Cells of MLNs and Spleen

MLN and spleen single-cell suspensions were prepared for flowcytometry analysis as previously described [61]. The cell suspensions were incubated with anti-mouse CD16/CD32 (Mouse BD Fc Block; BD Biosciences, Franklin Lake, NJ, USA) in PBS + 1% BSA for 15 min on ice to block unspecific binding sites. Afterwards, cells were stained with the following surface markers CD4-PerCp-Cy5.5, CD69-PE-Cy7, CXCR3-PE, CD25-AlexaFluor488, CD25-PE, (all purchased from eBioscience, San Diego, CA, USA) or T1ST2-FITC (MD Bioproducts, St. Paul, MN, USA) for 30 min on ice. Fixable Viability dye eFluor 780 (eBioscience) was used to exclude non-viable cells. Next, the cells were fixed and permeabilised with the Foxp3 Staining Buffer Set (eBioscience) according to the manufacturer’s protocol and then stained with the intracellular markers Foxp3-PE-Cy7, RORɣt-PE, IRF4-FITC, Tbet-eFluor660 or Gata3-eFluor660 (all purchased from eBioscience). Cells were measured on BD FACSCanto II flow cytometer, and results were analysed with FlowLogic software (Inivai Technologies, Mentone, Vic, Australia). The used gating strategy is shown in Appendix A.

### 2.9. Monoamine Levels in the Amygdala, Dorsal Hippocampus and Prefrontal Cortex

The levels of the monoamines noradrenaline (NA), dopamine (DA), 3,4-dihydroxyphenylacetic acid (DOPAC), homovanillic acid (HVA), 3-methoxytyramine (3-MT), serotonin (5-HT), 5-hydroxyindoleacetic acid (5-HIAA), and tryptophan (TRP) were measured in the amygdala, prefrontal cortex (PFC) and dorsal hippocampus (DH) by HPLC with electrochemical detection as previously described [60,62]. In these brain regions, 5-HT has a role in regulating anxiety and social behaviour [63,64]. Samples were pooled in pairs to reach detection minimum. Each sample contained two left brains (*n* = 4–5 per group). The frozen tissue samples were homogenised in an ice-cold solution containing 5 μM pargyline and 0.6 μM N-methylserotonin (NMET, internal standard). To 50 µL homogenate, 12.5 μL 2 M HClO_4_ was added, mixed, and placed in ice water. Thereafter, the homogenates were centrifuged for 15 min at 15,000× *g* (4 °C). The supernatants were diluted 10 times with mobile phase, which contained 50 mM citric acid, 50 mM phosphoric acid, 0.1 mM EDTA, 45 μL/L dibutyl amine, 77 mg/L 1-octanesulfonic acid sodium salt, 10% methanol. The pH of the buffer was adjusted to 3.4 with NaOH. The settings of the HPLC system were previously described in de Theije et al. [60]. Separation was performed at 40 °C using a flow rate of 0.8 mL/min. The concentration of each compound was calculated by comparison with both the internal and the external standards. The detection limit was 0.9 nM (signal/noise ratio 3:1). The turnovers were calculated by dividing the metabolite concentration by the monoamine concentration (5-HIAA/5-HT and (DOPAC + 3-MT + HVA)/DA).

### 2.10. Behavioural Tests

Anxiety-like and repetitive behaviours were assessed by the marble burying test as previously described [31,65]. Briefly, twenty black marbles were placed on the bedding in a cage (L35 × W20 × H15 cm) and the mice were placed individually. After 30 min, the number of marbles buried for 2/3 in the bedding was counted. Self-grooming was determined as previously described [66]. To measure self-grooming, in short, each mouse was placed individually in an empty cage for 10 min (first 5 min were considered as habituation period) and the cumulative time spent grooming and the frequency of grooming were analysed. Social behaviour was determined as previously described [66]. In brief, in an open field of L45 × W45 cm, a perforated plexiglass cage allowing visual and olfactory interaction was placed against a side. The mice were habituated for 5 min in the open field, followed by 5 min with an age- and sex-matched unfamiliar mouse in the plexiglass cage. Time in interaction zone and distance moved were analysed. The open field test was used to evaluate explorative behaviour, and the procedure was adapted from Seibenhener et al. [67,68]. The mice were individually placed in the centre of an open field (L45× W45 cm), cleaned with 70% alcohol, and recorded for 5 min with a Sony Handycam DCR-SR72 video camera. The time spent in the open field and locomotor activity were blindly analysed using the tracking software (Ethovision 3.1.16; Noldus, Wageningen, The Netherlands).

### 2.11. Statistical Analysis

Alpha-diversity indices (within-sample) and beta-diversity (between-sample) were used to examine changes in microbial community structures between control and different dietary mice samples. Alpha-diversity metrics (i.e., Shannon index, richness and evenness) were calculated from rarefied datasets (45,000 sequences/sample) using the package ‘vegan’ implemented in the R programming language (https://cran.r-project.org (accessed on 28 December 2021), https://github.com/vegandevs/vegan (accessed on 28 December 2021)). Differences in alpha-diversity indices between groups were assessed for significance using one-way analysis of variance (ANOVA) test and Sidak’s post hoc test. To examine beta-diversity differences in microbial community composition between samples, the pairwise Bray–Curtis dissimilarity (non-phylogenetic) metric was generated using the Primer7 software package and used to perform analysis of similarity (ANOSIM) calculations. ANOSIM was performed at the taxonomic level of genus, using square root transformed data, with 999 permutations, and data were visualised using non-multi-dimensional scaling (nMDS) incorporating taxa with strong Pearson’s correlation (R > 0.6).

Beta-diversity differences in relative abundance of individual taxa, between mice group samples, were assessed for significance using Kruskal–Wallis test controlling for false-discovery rate (FDR) using the Benjamini–Hochberg correction, implemented within the software package QIIME1.8. Taxa with an average abundance of (<1%) across the sample set were removed from the analysis. Furthermore, microbial relative abundances and Firmicutes-to-Bacteroidetes (F/B) ratios between conditions were studied. Additionally, an inferred 16S rRNA bacterial taxa model of the caecal content SCFA metabolite measurements were examined depicting the percent relative abundances of acetate, propionate, and butyrate-producing genera taxa [69,70], as well as the SCFAs and branch-chain fatty acids (BCFAs). Based on the Shaprio-Wilks normality test, these analyses used either the parametric one-way ANOVA and Sidak’s post hoc test or the non-parametric Kruskal–Wallis and Dunn’s post hoc tests.

Furthermore, statistical analysis was performed by comparing the control diet group to the enriched diets and by comparing the scGOS:lcFOS and the n-3 PUFA groups to the combination diet group. All data except the behavioural data were analysed using one-way ANOVA and Sidak’s multiple comparison post hoc test for selected comparisons. When not normally distributed or unequal variances, the data were transformed; if this failed, the non-parametric Kruskal–Wallis test was applied followed by Dunn’s multiple comparison post hoc test for selected comparisons. Marble burying, open field, self-grooming and social interaction were analysed using mixed models, controlled for repeated measures, litter effect and Sidak’s multiple comparisons test as post hoc analysis. Only grooming duration is not corrected for litter effect as the variance between dams is larger than the variance of the data. These data were considered statistically significant at *p* < 0.05.

### 2.12. Recursive Ensemble Feature Selection

In order to achieve insight into which microbial and immune features influence specific behaviours, we have analysed the whole dataset (38 samples and 87 features) regarding the marble burying (number of marbles buried) and open field (frequency in zone) tests at the age of 8 weeks (early adulthood). For marble burying data, we assigned the label ‘0’ for <10 and ‘1’ for ≥10 number of buried marbles. For the open field data, we assigned the label ‘0’ for ≥10 and ‘1’ for <10 number of entries into the zone. A feature selection algorithm for recursive ensemble feature selection (REFS) algorithm [71] was run based on the Borda Method [72]. In this algorithm, 8 different classifiers were used from the sci-kit learning toolbox [73]: Bagging, Random Forest, Logistic Regression, Gradient Boosting, Support Vector, Stochastic Gradient Descent, Passive Aggressive and Ridge. The algorithm assigns rankings to the features, depending on how they are used by the classifiers. Finally, this procedure was repeated in 10 cycles, where, at each run, we reduce the number of features to 80%.

## 3. Results

### 3.1. Caecum Microbiota Profiling and Short-Chain Fatty Acid Concentrations

Analysis of caecum content microbial communities at 8 weeks, using 16S ribosomal RNA gene amplicon sequencing, revealed that the microbial alpha-diversity indices were not significantly different between the control and three dietary groups, at the taxonomic level of genus (Shannon index: Figure 2A; evenness: Figure 2B). However, alpha-diversity was significantly higher in the combination diet mice compared to the scGOS:lcFOS mice (Shannon index (*p* < 0.05): Figure 2A) and (evenness (*p* < 0.01): Figure 2B). No significant differences in richness (observed species in a sample) between the four mice groups were observed (Figure 2C).

Significant differences in caecum content microbial community structure were observed between the control and each of the three dietary mice groups in beta-diversity analyses conducted on bacteria genera (ANOSIM (*p* = 0.001): Table 1). Additionally, the microbial communities differed between each dietary group (scGOS:lcFOS vs. combination, *p* = 0.002; and n-3 PUFA vs. combination, *p* = 0.001) (ANOSIM: Table 1). By incorporating Pearson’s correlation, this analysis indicated individual genera that are strongly associated (R > 0.6) to either the microbial communities of the control group or the three dietary groups (nMDS: Figure 3). These results indicate that scGOS:lcFOS, n-3 PUFAs and the combination significantly altered the intestinal microbiota profiles.

At the taxonomic level of phylum, the Firmicutes-to-Bacteroidetes ratio was significantly decreased (*p* < 0.05) in the combination diet group compared to the control group (Figure 2D). At the taxonomic level of genus, eight microbial features were significantly different (FDR-P ˂ 0.05) between groups, including *Allobaculum*, *S24-7* Unclassified, *Oscillospira*, *Ruminococcaceae* Unclassified, *Turicibacter*, *Akkermansia*, *Lachnospiraceae* Unclassified and *Rikenellaceae* Unclassified (Figure 4). In comparison to the control mice, the relative abundances of *Allobaculum* and *S24-7* Unclassified were higher (Figure 4A,B), and butyrate-producing *Oscillospira* lower in the scGOS:lcFOS mice (Figure 4C). Both the relative abundances of propionate-producing genera *Turicibacter* (Figure 4E) and *Akkermansia* (Figure 4F) were increased in n-3 PUFA mice, compared to control mice. Furthermore, the microbial composition in the combination diet mice indicated similar significant relative abundance bacterial alterations as scGOS:IcFOS, in comparison to the control mice. However, the combination diet mice also had a significant decrease in the relative abundances of acetate-producing *Ruminococcaceae* Unclassified (Figure 4D) and butyrate-producing *Lachnospiraceae* Unclassified (Figure 4G), with a significant relative abundance increase in propionate-producing *Akkermanisa* (Figure 4F) in comparison to the control mice.

Upon examining the microbial alterations between the dietary groups, n-3 PUFA mice had significantly higher relative abundances of acetate-producing *Ruminococcaeae* Unclassified, propionate-producing *Turicibacter*, and *Rikenellaceae* Unclassified, but lower relative abundance of propionate-producing *Akkermansia*, when compared to the combination diet mice (Figure 4D–F,H). Lastly, the combination diet mice had a significantly higher relative abundance of propionate-producing *Akkermansia*, assessed to the scGOS:IcFOS mice (Figure 4F).

Intestinal bacteria ferment carbohydrates and proteins into the different SCFAs and BCFAs. Using our inferred SCFA metabolite 16S rRNA bacterial taxa model, the percent relative abundances of putative bacteria that ferment carbohydrates and produce acetate, propionate and butyrate differed between the control and dietary mice groups. The genera examined as inferred SCFA-producing bacterial metabolites included acetate (*Ruminococcaceae* Unclassified, *Lactobacillus*, *Ruminococcus*, *Parabacteroidetes*, *Dorea*, *Streptococcus*, and *Bifidobacterium*); propionate (*Bacteroides*, *Akkermansia*, *Turicibacter*, and *Prevotella*); and butyrate (*Lachnospiraceae* Unclassified, [*Ruminococcus*], *Oscillospira*, *Lachnospiraceae* Other, *Coprococcus*, *Roseburia*, *Anaerofustis*, *Butyrivibrio*, *Anaerostipes*, and *Anaerotruncus*).

The combination diet mice significantly lowered the percent relative abundance of putative genera acetate-producing bacteria, compared to the n-3 PUFA mice (Figure 5A). Additionally, the combination diet mice had a significantly higher relative abundance of putative genera propionate-producing bacteria than the control mice (Figure 5B). Finally, only the n-3 PUFA mice significantly lowered the percent relative abundance of putative genera butyrate-producing bacteria compared to the control mice (Figure 5C). Interestingly, these inferred bacterial genera metabolomics data were approximately mirrored by the actual targeted SCFA metabolomics concentrations levels in the caecum content.

Next, the targeted SCFA and BCFA metabolomics concentrations were examined using gas chromatography (Figure 5D–I). When compared to the control mice, the caecum content total SCFA and propionate concentration levels were significantly increased in the scGOS:lcFOS mice (Figure 5D,F). Additionally, the propionate concentration level was significantly increased in the combination diet mice, compared to the control mice (Figure 5F). Although not significantly different across all four mice groups, the scGOS:lcFOS mice concentration levels suggest that both acetate and butyrate are trending higher, compared to the other mice groups (Figure 5E,G). Finally, the concentration levels of the BCFAs for valerate, isobutyrate and isovalerate were affected by the different diets, with a significant decrease in valerate, iso-butyrate, and iso-valerate in the combination mice, compared to the n-3 PUFA mice (Figure 5H–J).

### 3.2. Intestinal Serotonin-Producing Cells

Serotonin is a key neurotransmitter in the bidirectionally communication between the enteric and the central nervous system. To assess the effect of the diets on the intestinal serotonin-producing cells, the number of enterochromaffin cells was measured in the intestine. The number of these cells was significantly increased in the jejunum of the mice receiving a diet supplemented with scGOS:lcFOS, n-3 PUFAs or the combination compared to mice receiving the control diet (Figure 6A). The number of enterochromaffin cells in the ileum was unaffected by the different diets (Figure 6B). The cell number in the colon was significantly lower in the combination diet group than in the n-3 PUFA group. However, no differences were observed compared to the control (Figure 6C). Figure 6 contains representative pictures of 5HT + enterochromaffin cells the jejunum (D), ileum (E) and colon (F) of control mice.

### 3.3. Immune Modulation

The n-3 PUFA diet significantly increased the percentage of total activated T cells (CD69^+^ of CD4^+^ cells) compared to the control group in the MLNs (Figure 7A). This n-3 PUFA-induced increase was not due to more activated Th1 or Th2 cells in the MLN (Appendix A). In the MLN, the Th1 (CXCR3^+^ CD4^+^) and the Th2 (T1ST2^+^ CD4^+^) cell response tended towards an increase in, respectively, the scGOS:lcFOS group and the n-3 PUFA group compared to the combination diet (Figure 7B,C). Representative flowcytometry plots and histograms are shown in Appendix A. These analyses were also performed in the spleen, but no significant differences were observed (Appendix A–G).

In this study, the percentage of Tregs (CD25^+^ Foxp3^+^ CD4^+^) was unaltered by the dietary interventions in the MLN as well as in the spleen (Appendix A). Additionally, the percentage Th17 cells (RORγt^+^ CD4^+^) was unaffected by the different diets (Appendix A).

### 3.4. Monoamines Levels in Brain Regions

Several monoamines were measured in the amygdala, prefrontal cortex and hippocampus. In the amygdala, the 5-HT (serotonin) and the 5-HIAA (serotonin metabolite) levels were significantly increased in the combination diet group compared to the scGOS:lcFOS group (Figure 8B,C). However, tryptophan levels as well as serotonin turnover were not significantly changed by the different diets compared to the control diet (Figure 8A,D). Except for the level of DOPAC, which tended towards an increase in the combination diet group compared to the n-3 PUFA group, the levels of noradrenaline, dopamine and the dopamine metabolites were unaltered as well as the turnover of dopamine (Appendix A). In the prefrontal cortex and dorsal hippocampus, no significant effects of the different diets were observed (Appendix A, respectively).

### 3.5. Behavioural Modulation

During adolescence and early adulthood anxiety-like, self-grooming, explorative and social behaviour were assessed. 

#### 3.5.1. Marble Burying and Self-Grooming

Overall, the number of marbles buried was unaffected by diet (*F*
_(3, 10.224)_ = 1.640, *p* > 0.05) but tended towards an effect by age (*F*
_(1, 34)_ = 3.323, *p* = 0.077), and the change in number of buried marbles over time was independent of the diet (interaction effect between diet and age (*F*
_(3, 34)_ = 2.198, *p* > 0.05)) (Figure 9A). The mice receiving the control diet buried significantly more marbles in early adulthood compared to adolescence (*p* < 0.05). The number of buried marbles in the dietary groups was unchanged over time. During adolescence the number of buried marbles was unaffected by the diets. However, in early adulthood the mice receiving the scGOS:lcFOS diet tended to bury less marbles compared to the control group (*p* = 0.095). One might be doubtful that the number of buried marbled is not significantly different between the control and scGOS:lcFOS group in early adulthood. These data are controlled for litter effect and can lead to the fact that visuals and statistics are not completely aligned.

Grooming duration was overall affected by age (*F*
_(1, 33.381)_ = 6.502, *p* < 0.05) and diet (*F*
_(3, 33.222)_ = 3.387, *p* < 0.05). The change in grooming duration over time was independent of the diet (*F*
_(3, 33.388)_ = 0.629, *p >* 0.05) (Appendix A). Grooming duration tended towards a decrease over time in the control (*p* = 0.079) as well as in the scGOS:lcFOS group (*p* = 0.058). In early adulthood, grooming duration showed an increasing trend in the combination group compared with the scGOS:lcFOS group (*p* = 0.074). The frequency of grooming was neither affected by age nor diet (Appendix A).

#### 3.5.2. Open Field

The explorative behaviour was measured by frequency and time in the centre of the open field. The frequency in the centre was significantly affected by age (*F*
_(1, 34)_ = 12.558, *p <* 0.01), but not by diet (*F*
_(3, 10.935)_ = 1.519, *p* > 0.05) and the frequency in the centre over time was dependent on the diet (interaction effect between diet and age (*F*
_(3, 34)_ = 5.820, *p* < 0.01)) (Figure 9B). Over time, the number of entries was significantly reduced in the control group and in the scGOS:lcFOS group (*p* < 0.001, *p* < 0.05, respectively). In adolescence, the number of entries tended towards a reduction in the combination diet group compared to the scGOS:lcFOS group (*p* = 0.088). The time spent in the centre of the open field tended towards an overall effect by age (*F*
_(1, 34)_ = 3.025, *p* = 0.0951) but no overall effect by diet (*F*
_(3, 11.042)_ = 1.409, *p* > 0.05). The time spent in the centre over time depended on the diet (interaction effect between diet and age (*F*
_(3, 34)_ = 4.414, *p* < 0.05)). The statistical effects for time in centre were similar to frequency in centre (Figure 9C). Locomotor activity (total distance moved) also showed the same pattern (Age: *F*
_(1, 34)_ = 2.932, *p =* 0.096. diet: *F*
_(3, 11.762)_ = 1.545, *p* > 0.05, interaction: *F*
_(3, 34)_ = 3.828, *p* < 0.05) (Appendix A).

#### 3.5.3. Social Interaction

Social interaction, shown as relative time in zone (target/no target), was neither affected by age (*F*
_(1, 32.933)_ = 0.997, *p* > 0.05) nor diet (*F*
_(3, 8.300)_ = 0.576, *p >* 0.05) (interaction (*F*
_(3, 32.827)_ = 0.086, *p* > 0.05)) (Appendix A). However, overall, locomotor activity (distance moved) was significantly affected by age and unaffected by diet (Age: *F*
_(1, 34)_ = 7.425, *p <* 0.05, diet: *F*
_(3, 34)_ = 1.231, *p >* 0.05, interaction: *F*
_(3, 34)_ = 2.343, *p* > 0.05). Locomotor activity in the control and scGOS:lcFOS-receiving mice was significantly decreased over time (*p* < 0.05 for both groups) (Appendix A). In adolescence, locomotor activity tended towards a reduction in the combination diet group compared to the scGOS:lcFOS group (*p* = 0.076).

### 3.6. Feature Selection Regarding Repetitive and Explorative Behaviour

Though the effects of the different early-life dietary interventions on repetitive and explorative behaviour of healthy adolescent and early adult mice are not very clear, we have analysed the whole dataset regarding marble burying (anxiety-like and repetitive behaviour) and open field (explorative behaviour) tests in early adulthood to achieve insight into which microbial, immune and monoamine features influence the two behavioural outcomes independent on the dietary intervention. After running the REFS algorithm 10 times, for marble burying behaviour, the best signature is at 9 features, with an average global accuracy of all classifiers of 0.73 (Figure 10A). The optimal associated receiver-operating characteristic (ROC) curve for the best-performing classifier Ridge is shown in Figure 10B that demonstrates an area under the curve (AUC) of 0.84 ± 0.17, which is considered good to outstanding [74,75]. The magnitudes of the 9 features separating the two groups are presented in Figure 10C. In Table 2, the direction of change comparing the groups ≥10 (label 1) with <10 (label 0) number of buried marbles of the 9 features are presented. A reduction in marble burying (associated with reduced repetitive or anxiety-like behaviour) was associated with reductions in the caecal content relative abundance of the genus *Adlercreutzia*, the alpha-diversity Shannon index, and tryptophan levels in dorsal hippocampus; upregulation of the caecal content relative abundance of the genus *Dehalobacterium*, percentages of Th1 (CXCR3^+^ CD4^+^) and Th17 (RORγt^+^ CD4^+^) cells in MLN, and percentage of activated Th2 splenocytes (CD69^+^ T1ST2^+^ CD4^+^) (Figure 10C; Table 2). An overview of the involved features and their connection is depicted in Figure 11A.

For open field behaviour, after running the REFS algorithm 10 times, the best signature is at 16 features with average accuracy of all classifiers of 0.77 (Figure 12A). The ROC curve for the best-performing classifier Support Vector is shown in Figure 12B that demonstrates an area under the curve (AUC) of 0.82 ± 0.20. The magnitudes of 16 features separating the two groups are presented in Figure 12C. In Table 3, the direction of change comparing the groups ≥10 (label 0) with <10 (label 1) number of entries into the centre of the open field of 16 features is presented. Increased entry into the centre of the open field (more explorative behaviour) is associated with an increase in the caecal content relative abundance of the genera *Odoribacter* and *Turibacter*, dopamine and serotonin turnover in the amygdala and prefrontal cortex, respectively, and 5-HIAA levels in the prefrontal cortex. A reduction in the caecal content relative abundances of the phyla Cyanobacteria (class 4C0d; order YS2; unclassified family and genus), genera *Oscillospira*, *Ruminococcus*, *Lachnospiraceae* Other;Unclassified and *Adlercreutzia* as well as reduced levels of HVA in the prefrontal cortex are associated with enhanced explorative behaviour (Figure 12C; Table 3). An overview of the involved features and their connection is depicted in Figure 11B.

## 4. Discussion

The present study demonstrates that dietary supplementation with the combination of scGOS:lcFOS and n-3 PUFAs leads to a distinct caecal content microbiota composition and indicates a balanced immune response compared to the individual food components. In our previous work [31], scGOS:lcFOS effects on the in intestinal microbiota were associated with improved social behaviour and reduced anxiety-like and stereotypic behaviour assessed by marble burying and self-grooming behavioural tests. In this study, the effects of scGOS:lcFOS on these behaviours are less clear.

All three diets modulated caecal content microbial community structures. The distinct caecal content microbial profile alterations were significant comparing the control mice to the three dietary mice groups. Further, caecal content bacterial composition in the combination group was different from both scGOS:lcFOS and n-3 PUFAs. These results indicate that the NDO diets modulated the caecal content microbial community structures uniquely, which is in line with other studies [23,24,31,38]. Based on the relative abundances of individual bacteria at the taxonomic level of genus, the combination diet induced a microbial profile similar to the profile induced by scGOS:lcFOS. Interestingly, scGOS:lcFOS and the combination diet interventions increased the relative abundances of genera *Allobaculum* and *S24-7* Unclassified, while reducing the abundance of the genera *Oscillospira* and *Ruminococcaceae* Unclassified. The genera *Allobaculum* and *S24-7* Unclassified have both been reported to be involved in the fermentation of fibres and to be putative SCFA-producing bacteria [31,76,77,78]. The genus *Oscillospira* has been linked to slow faecal transit, which could lead to more water absorption from the stool and eventually result in constipation. scGOS:lcFOS has been reported to affect stool consistency, decrease transit time [79] and increase defecation frequency [80]. Thus, these functions of scGOS:lcFOS could be due to the lower relative abundance of the genus *Oscillospira*. The relative abundance of the genus *Ruminococcaceae* Unclassified has been described to be positively correlated with plasma levels of serotonin [81]. This might indicate that taxa associated with the family of Ruminococcaceae could play a role in tryptophan metabolism, converting tryptophan into serotonin [82].

To date, intestinal microbiota modulation by n-3 PUFAs has been less defined. In this study, the n-3 PUFA diet induced changes to the relative abundances of the genera *Turicibacter* and *Akkermansia*. Since *Turicibacter* has been shown to enhance the levels of poly-unsaturated fatty acids [83], this could be a positive feedback mechanism, where enhanced availability of lipids such as n-3 PUFAs increased the abundance of *Turibacter*. *Akkermansia* is induced by n-3 PUFAs, as mentioned above, but *Akkermansia* is induced significantly more by the combination diet group, indicating an additive effect on the induction of *Akkermansia* when scGOS:lcFOS and n-3 PUFAs are combined. *Akkermansia* has been reported to be a propionate-producing bacteria [84] which could explain the increased caecal content propionate levels in the combination diet group. *Akkermansia* has been reported to communicate with the immune system of the host and, i.e., induce regulatory T cells [85] and reduce inflammation [86]. With regard to the F:B ratio, an increase in this ratio has been associated with high-fat diets (Westernised diets) [87] and obesity [88]. Moreover, the F:B ratio in mice receiving a n-3 PUFA-rich diet was decreased [89], which is in line with our findings, indicating a positive effect on the F:B ratio by scGOS:lcFOS and n-3 PUFAs.

As expected and observed before [31], the absolute SCFA levels in the scGOS:lcFOS group were increased indicating a saccharolytic fermentation profile. We expected the effects of scGOS:lcFOS on the SCFA levels to be visible in the combination group, however, that was not the case, except for propionate. Both caecal content inferred propionate-producing genera and propionate metabolite measurements were significantly enhanced in the combination diet group, compared to the control group. Based on the acetate producers, one would expect less acetate in the combination diet group than in the n-3 PUFA group, but the acetate levels in these groups were similar. This could be due to overlapping function of other bacteria compensating for decreased relative abundance of acetate-producing bacteria in the combination group. Concerning SCFA levels, this might indicate that the n-3 PUFAs in some way influence the fermentation of scGOS:lcFOS, which leads to less pronounced effects in SCFA levels in the combination diet group. The other possibility is the impact of n-3 PUFAs and scGOS:lcFOS on intestinal epithelial cell uptake of SCFA. If n-3 PUFAs and scGOS:lcFOS improve intestinal epithelial function and increase their ability to uptake more SCFAs, then luminal SCFA levels would remain unchanged in spite of increased production by the SCFA-producing bacteria.

SCFAs and intestinal bacteria have the capacity to influence the immune system [90,91]. Therefore, modulation of intestinal microbiota composition and/or activity might eventually lead to immune modulation. SCFAs can act both pro-and anti-inflammatory dependending on the kind and state of the immune cell [90]. They influence the immune system through inhibition of HDAC activity and the GPCRs GPR41, GPR43 and GPR109. Among others, SCFAs induce genes that maintain intestinal barrier function and induce differentiation and function of T cell subsets into Th1, Th17 and Tregs [92]. These intestinal microbial and SCFA data indicate that as, individual components, scGOS:lcFOS and n-3 PUFAs influence the intestinal microbiota and it seems that the combination of scGOS:lcFOS and n-3 PUFAs balances intestinal microbiota composition—the best of both.

Previous studies have shown that scGOS:lcFOS is able to induce a Th1 response [93,94] and stimulate the secretion of IL-10 from DCs, which eventually results in upregulation of the number of suppressive Tregs [34,35]. Although in this study the T cell subsets were unaffected by scGOS:lcFOS and n-3 PUFAs, Th1 and Th2 response tended towards a decrease in the combination diet group compared with scGOS:lcFOS and n-3 PUFAs, respectively. Considering that this study takes place in healthy mice, one could argue that a pronounced immune response is undesired. Therefore, these data might indicate that the combination of scGOS:lcFOS and n-3 PUFAs leads to a balanced immune response. Both scGOS:lcFOS and n-3 PUFAs can induce Tregs. A reduced Treg response plays an essential role in preventing diseases such as food allergies [35,44]. It is most likely considered acceptable that the percentages of Tregs and Th17 were unaffected by the dietary interventions as significant changes may be undesired in a healthy host. N-3 PUFAs are known to be health promoting because of their anti-inflammatory potential. However, in this study, n-3 PUFAs induced the activated CD4^+^ cells in the MLN, which was not in line with other studies [44,95]. Since the n-3 PUFA-induced increase in CD69 ^+^ CD4^+^ cells was not caused by enhanced activation of Th1 or Th2 cells, another explanation could be that n-3 PUFAs induce memory T cells, as CD69 can also be expressed by these cells [96]. However, in the combination diet group, this increase in activated CD4^+^ cells was no longer observed. This might indicate that the induction of activated CD4^+^ cells is hampered by scGOS:lcFOS. All the immunological findings were only observed in the MLN. The diets had no pronounced effects in the spleen, suggesting that the diets might exert their modulatory effects locally in the intestine.

As mentioned earlier, n-3 PUFAs and scGOS:lcFOS can modulate the intestinal microbiota, the immune system, and also brain development and behaviour [25,31,38,40,45,46,48]. Overall, the change in behavioural parameters in this study mainly occurred over time: less repetitive and explorative behaviour and more anxiety-like behaviour. The scGOS:lcFOS behavioural effects are in line with previous studies, where scGOS:lcFOS modulated repetitive and anxiety-like behaviour in healthy mice [31,38]. Additionally, in a mouse model of stress-induced anxiety, prebiotics improved or prevented the anxiety-like behaviour (assessed by light/dark preference and open field tests) [97]. n-3 PUFAs have been shown to improve anxiety-like and social behaviour in healthy and allergic rodents, respectively [47,48]; this is not in line with our observations. However, Robertson et al. reported no effects of n-3 PUFAs on repetitive and anxiety-related behaviour (assessed by marble burying, light/dark preference and elevated plus maze tests) in healthy mice [25] and this matches our data. Robertson et al. also reported the importance of n-3 PUFAs; n-3 PUFA deficiency in healthy mice led to impaired behaviour in adolescence and later in life [25]. This indicates that n-3 PUFAs are an essential dietary component; but to observe behavioural improvements, maybe the window of opportunity is too small in a healthy host.

Behavioural changes are often accompanied by alterations in neurochemical mediators such as serotonin in the brain. Serotonin is a metabolite from tryptophan, and tryptophan is an essential amino acid that needs to be obtained from the diet. Serotonin is able to modulate anxiety and social behaviour [64,98]. Although not significantly lower, tryptophan levels in the brain regions investigated in this study show a similar pattern. Tryptophan levels seemed lower in the scGOS:lcFOS group and might indicate a lower availability of tryptophan. As tryptophan enters the body through the stomach and intestine, it could be that tryptophan-utilising intestinal bacteria use some of the tryptophan. Another option is that serotonin production by specific epithelial cells called enterochromaffin cells increases by scGOS:lcFOS stimulation. Indeed, the number of serotonin-positive cells in the jejunum was significantly increased compared with the control group. For this boost of serotonin secretion, tryptophan is essential and consequently less tryptophan is available for the brain. Intriguingly, this phenomenon was only observed in the scGOS:lcFOS group. All three dietary interventions increased serotonin-positive cells in the jejunum, but no changes in tryptophan levels in the brain were observed in the combination and n-3 PUFA diet groups compared to the control group. The reason for this discrepancy remains unclear. Enterochromaffin cells are located throughout the intestinal tract [99], and the reason that the dietary effect is only observed in the jejunum and not in the ileum and colon is unknown. The analysis of goblet and Paneth cell counts would provide a more complete picture; however, these analyses are missing in this study. Intestinal bacteria associated with the family Ruminococcaceae possibly play an additional role. A recent study showed that the enhanced abundance of the family Ruminococcaceae correlated significantly with increased 5-HT_2A_ receptor density in the PFC in a maternal activation (MIA) murine model [100]. The fact that MIA offspring mice show impaired behaviour and that the abundance of faecal family Ruminococcaceae is increased in young children diagnosed with autism spectrum disorder might explain why scGOS:lcFOS improves behaviour via a reduction in Ruminococcaceae in the intestines of healthy mice [31,100,101,102]. More studies are needed to elucidate a more detailed mechanism.

Serotonin and serotonin metabolite levels in the investigated brain regions approximately follow the same pattern in the dietary groups. The combination diet group increased serotonin and serotonin metabolite levels compared with the scGOS:lcFOS group in the amygdala brain region. In general, more serotonin is desirable according to the literature. However, in our previous paper, scGOS:lcFOS led to improved behaviour, but lower serotonin levels [31]. Noticeable, in this study, we observed no significant differences between the dietary groups in the other monoamines. The monoamines do follow the same pattern; the levels in the scGOS:lcFOS and n-3 PUFA groups are lower than in the control and the levels in the combination diet group approximated to the levels in control. This again indicates that the combination diet balances the effects of scGOS:lcFOS and n-3 PUFAs, which might be of valuable interest when combining these components in a healthy host or in models of disease. The main impact of n-3 PUFAs in this study was the modulated intestinal microbiota, which match finding of other studies [89,103]. To be able to explain the results, data on the n-3 PUFA metabolites in blood and/or brain tissues might have been useful. Overall, regarding the effects of scGOS:lcFOS with or without n-3 PUFAs on microbial composition and monoamine levels in different brain areas, it can be hypothesised that scGOS:lcFOS is influencing explorative behaviour either directly in the brain or indirectly through the microbe–brain axis. Future studies are essential to further elaborate on these hypotheses.

Finally, we implemented the REFS algorithm on all microbial, immune (MLN and spleen) and monoamine (brain) data and found 8 and 16 features, whose changes significantly seem to predict changes in repetitive and explorative behaviour with a global accuracy of 73% and 77%, respectively. Predictive changes for repetitive behaviour are mostly influenced by the composition of certain intestinal bacteria, which might in turn affect local and system immune balance, resulting in modified tryptophan levels in the dorsal hippocampus. Comparable to our findings, low hippocampal tryptophan levels are associated with an anxiolytic effect in BALB/c mice [104]. A previous study has shown that enhanced intestinal abundance of *Adlercreutzia* is associated with inflammation-induced depressive-like behaviour in mice [105]. Increased expression of the genus *Dehalobacterium* in mice is associated with ageing and food intervention-induced anti-inflammatory effects [106,107]. Moreover, reduced levels of intestinal *Dehalobacterium* observed in BTBR mice that have an autistic phenotype are associated with enhanced marble burying [108]. Taken together, our finding that reduced marble burying is associated with reduced *Adlercreutzia* and increased *Dehalobacterium* abundance seems to be compatible with previous murine behavioural studies. Unlike repetitive behaviour, explorative behaviour was not associated with either local or systemic T lymphocytes, but was significantly associated with microbial alterations in the relative abundances of several bacterial genera plus changes in serotonin and dopamine metabolism in PFC and amygdala. No specific reports on the role of the genera *Oscillospira*, *Ruminococcus, Odoribacter*, *Turicibacter*, *Lachnospiraceae* Other/Unclassified and *Adlercreutzia* on explorative behaviour are published [109].

In conclusion, both early-life dietary interventions with scGOS:lcFOS and/or n-3 PUFAs affected caecal content microbial profiles, but had limited effects on behaviour and the immune system. No apparent additive effect was observed when scGOS:lcFOS and n-3 PUFAs were combined, as the data from the combination diet group show the same pattern as scGOS:lcFOS for some parameters and the n-3 PUFA pattern for other parameters. All parameters considered, the results suggest that these dietary components together create a balance—the best of both in a healthy host. The limited effect on the immune system and behaviour is considered acceptable as this study was carried out in healthy mice. It may be concluded that improving intestinal microbiota composition by diet in a healthy host, where SCFA production by bacteria and intestinal epithelial cells is normal, has no functional impact. However, in a diseased situation (i.e., allergy, colitis or neurological disease), where the microbiota is abnormal, improving the microbiota by diet might have a positive impact.

## Figures and Tables

**Figure 1 nutrients-14-00173-f001:**
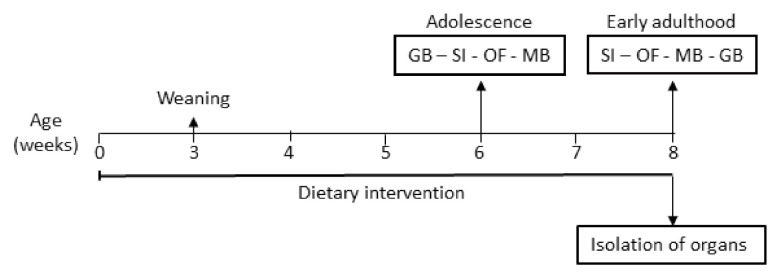
Schematic overview of the experimental protocol and the executed behavioural tests. From day of birth, the dams received a control diet, a 3% scGOS:lcFOS (9:1)-enriched diet, a n-3 PUFA diet or a combination diet containing 3% scGOS:lcFOS and n-3 PUFAs. The pups were weaned when 3 weeks old and continued on the allocated diet to the end of the experiment. During adolescence and early adulthood, a battery of behavioural tests was conducted. Organs were collected after the last behavioural test. GB: grooming behaviour, SI: social interaction test, OF: open field test, and MB: marble burying test.

**Figure 2 nutrients-14-00173-f002:**
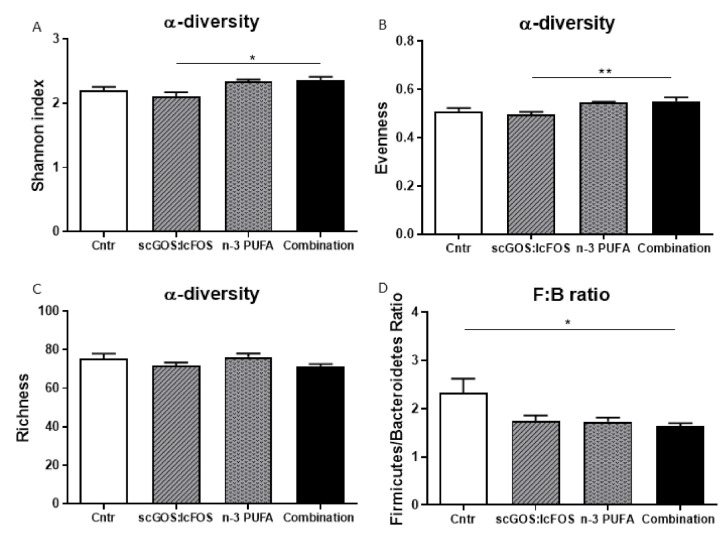
Caecum content alpha-diversity and Firmicutes-to-Bacteroidetes ratio for the control and dietary mice groups. Alpha-diversity indices were examined at the taxonomic level of genus. Alpha-diversity indices rarefied to 45,000 sequences per sample. Analysis of the (**A**) Shannon index and (**B**) evenness both indicated a significant dietary effect across all groups, with the combination diet diversity significantly higher than scGOS1cFOS. (**C**) Richness diversity was not significantly different across groups. At the taxonomic level of phylum, the (**D**) Firmicutes-to-Bacteroidetes ratio significantly decreased in the combination diet mice compared to the control mice. Data were square root transformed for statistics. (**A**–**D**): Data shown as the mean +/− SEM. Analysed by one-way ANOVA and Sidak’s multiple comparisons post hoc test. * *p* < 0.05, ** *p* < 0.01. *n* = 8–10 mice per group.

**Figure 3 nutrients-14-00173-f003:**
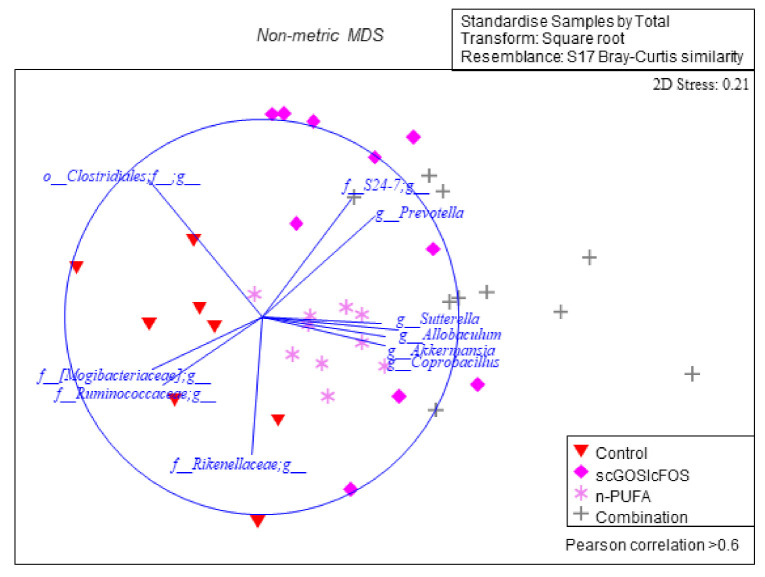
Visual of the non-multi-dimensional scaling (nMDS) plot depicting caecum content microbial community structures between control and dietary mice samples. A significant microbial community structure was observed between the control mice and the three dietary treatments. For statistical details, reference the analysis of similarity (ANOSIM) calculations in Table 1. Identified taxa with Pearson’s correlation (>0.6) were strongly associated with either the control or dietary interventions.

**Figure 4 nutrients-14-00173-f004:**
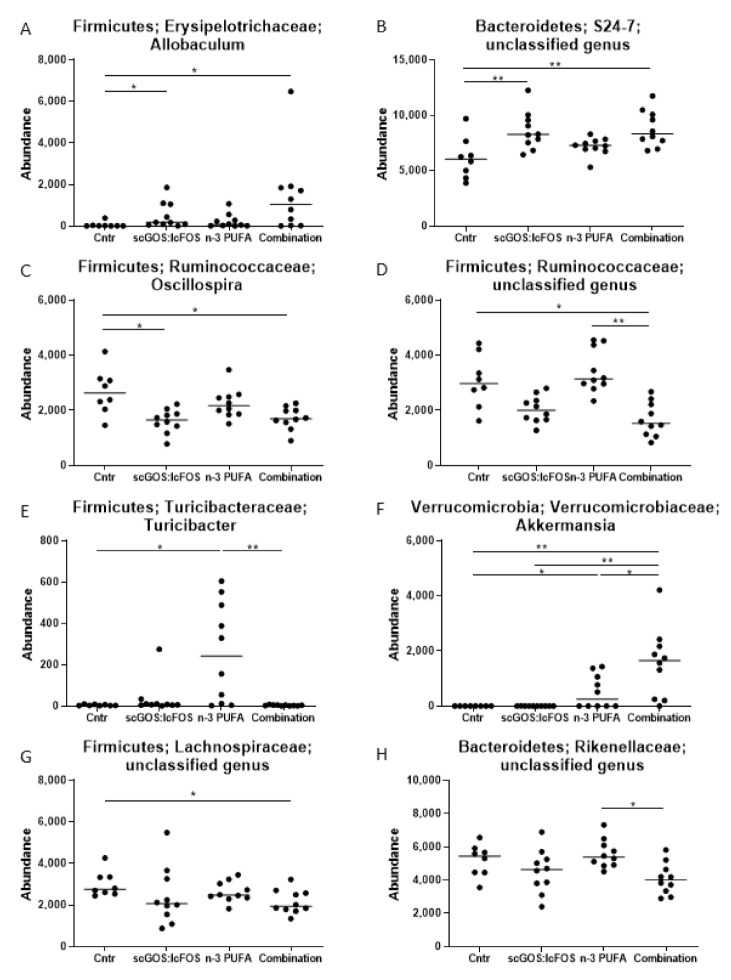
Significant genera taxa-specific relative abundances across mice groups (caecum content). The abundances of eight significant genera across mice groups are identified as: (**A**) Allobaculum, (**B**) S24-7 Unclassified, (**C**) Oscillospira, (**D**) Ruminococcaceae Unclassified, (**E**) Turicibacter, (**F**) Akkermanisa, (**G**) Lachnospiraceae Unclassified, and (**H**) Rikenellaceae Unclassified. (**A**–**H**): Data shown as individual data points and median. Assessed for significance using Kruskal–Wallis test controlling for false-discovery rate (FDR): * FDR-P < 0.05, ** FDR-P < 0.01. *n* = 8–10 mice per group.

**Figure 5 nutrients-14-00173-f005:**
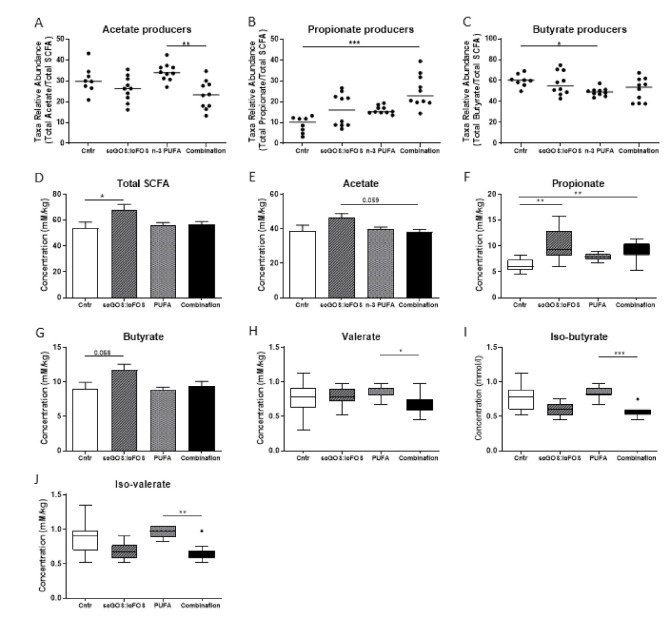
Caecum content predicted and targeted short-chain fatty acids metabolite concentrations between mice groups. Genera taxa metabolite predictive models depicting the relative abundances of (**A**) acetate- (**B**) propionate- and (**C**) butyrate-producing taxa were examined in the control and dietary mice groups. Targeted SCFA and BCFA graphs depict (**D**) total SCFA (mM/kg); (**E**) acetate (mM/kg); (**F**) propionate (mM/kg); (**G**) butyrate (mM/kg); (**H**) valerate (mM/kg); (**I**) iso-butyrate (mM/kg) and (**J**) iso-valeric acid (mM/kg) in the mice groups. (**A**–**C**): Data shown as individual data points and median. (**D**,**E**,**G**): Data shown as the mean +/− SEM. (**F**,**H**–**J**): Data shown as box-and-whiskers Tukey plots. (**A**–**C**,**F**,**H**–**J**): Analysed by Kruskal–Wallis and Dunn’s multiple comparisons post hoc tests. (**D**,**E**,**F**): Analysed by one-way ANOVA and Sidak’s multiple comparisons post hoc test. * *p* < 0.05, ** *p* < 0.01, *** *p* < 0.001. (**A**–**J**): *n* = 8–10 mice per group. SCFA: short-chain fatty acids.

**Figure 6 nutrients-14-00173-f006:**
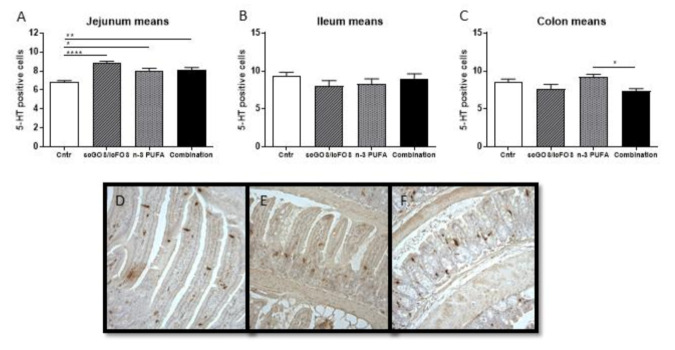
Serotonin-producing cells in the jejunum, ileum and colon. (**A**) In the jejunum, the number of serotonin-producing cells increased in the scGOS:lcFOS, n-3 PUFA and the combination diet groups compared to control. (**B**) In the ileum, the diets did not affect the serotonin-producing cells. (**C**) In the colon, the number of serotonin-producing cells was significantly decreased in the combination diet group compared to the n-3 PUFA group. (**D**–**F**) Representative pictures (control group) of the jejunum (**D**), ileum (**E**) and colon (**F**). (**A**–**C**): Data shown as the mean +/− SEM. Analysed by one-way ANOVA and Sidak’s multiple comparisons post hoc test. * *p* < 0.05, ** *p* < 0.01, **** *p* < 0.0001. *n* = 6 mice per group.

**Figure 7 nutrients-14-00173-f007:**
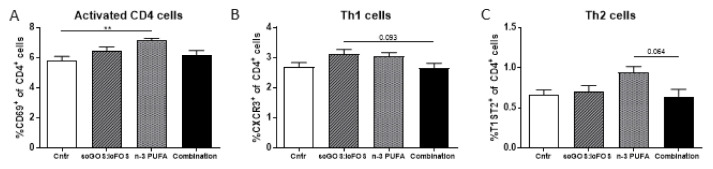
The dietary effect of scGOS:lcFOS, n-3 PUFAs and the combination of scGOS:lcFOS and n-3 PUFAs on activated CD4 cells, Th1 cells and Th2 cells in the MLN. (**A**) The percentage of activated CD4 (CD69^+^ CD4^+^) cells is significantly increased in the n-3 PUFA group compared to control. (**B**) The percentage of Th1 cells (CXCR3^+^ CD4^+^) tended to a decrease in the combination diet group compared to the scGOS:lcFOS group. (**C**) The percentage of Th2 cells (T1ST2^+^ CD4^+^) tended to a decrease in the combination diet group compared to the n-3 PUFA group. An outlier in the control group was excluded by use of ROUT analysis. (**A**–**C**): Data shown as the mean +/− SEM. Analysed by one-way ANOVA and Sidak’s multiple comparisons post hoc test. ** *p* < 0.01. *n* = 5–10 mice per group, 2 samples in the control group, 3 samples in the n-3 PUFA group and 2 samples in the combination diet group were excluded due to low number of viable cells. Th1: T helper 1 cells. Th2: T helper 2 cells.

**Figure 8 nutrients-14-00173-f008:**
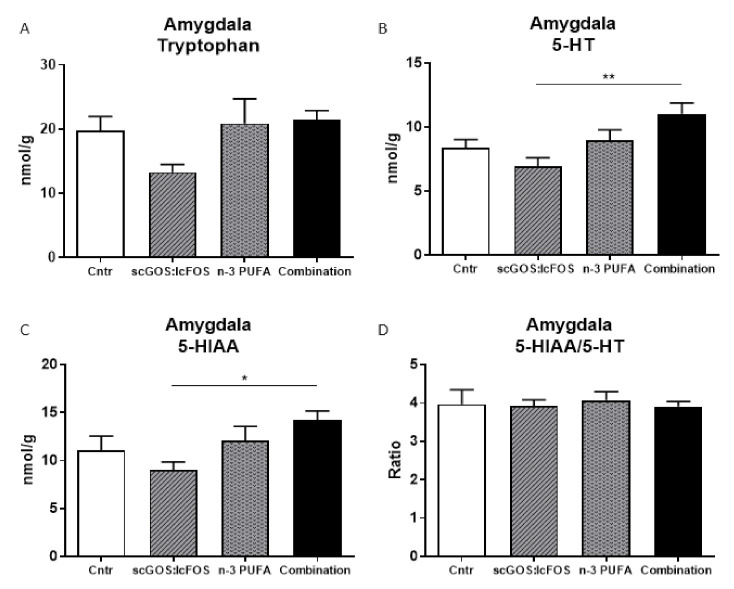
The amygdala levels of tryptophan, 5-HT, 5-HIAA and the serotonin turnover. (**A**) The tryptophan level did not differ between the dietary groups. (**B**,**C**) The 5-HT and 5-HIAA levels were significantly increased in the combination diet group compared to the scGOS:lcFOS group. (**D**) The serotonin turnover was unchanged in the dietary groups. (**A**–**D**): Data shown as the mean +/− SEM. Analysed by one-way ANOVA and Sidak’s multiple comparisons post hoc test. * *p* < 0.05, ** *p* < 0.01. *n* = 4–5 samples per group, samples were pooled in pairs, in order to reach detection minimum, each sample contained two left brains. 5-HT: serotonin. 5-HIAA: 5-hydroxyindoleacetic acid (serotonin metabolite).

**Figure 9 nutrients-14-00173-f009:**
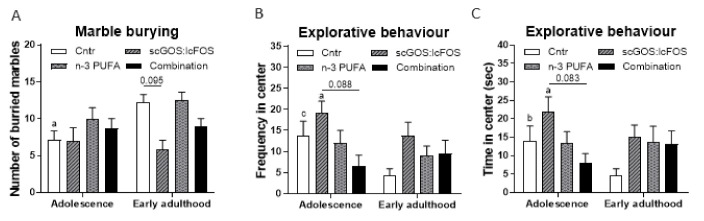
Anxiety-like behaviour assessed by marble burying and explorative behaviour in the open field. (**A**) Number of buried marbles by mice receiving the control, scGOS:lcFOS, n-3 PUFA or combination diet in adolescence and early adulthood (**B**) Explorative behaviour, the frequency the mice receiving the control, scGOS:lcFOS, n-3 PUFA or combination diet entered the centre of the open field in adolescence and early adulthood. (**C**) Explorative behaviour, the time the mice receiving the control, scGOS:lcFOS, n-3 PUFA or combination diet spent in the centre of the open field. (**A**–**C**): Data shown as the mean +/− SEM. Analysed with mixed models, controlled for repeated measures, litter effect and Sidak’s multiple comparisons post hoc test. a = * *p* < 0.05 compared with early adulthood within diet group, b = ** *p* < 0.01 compared with early adulthood within diet group, c = *** *p* < 0.001 compared with early adulthood within diet group. (**A**–**C**): *n* = 8–10 mice per group.

**Figure 10 nutrients-14-00173-f010:**
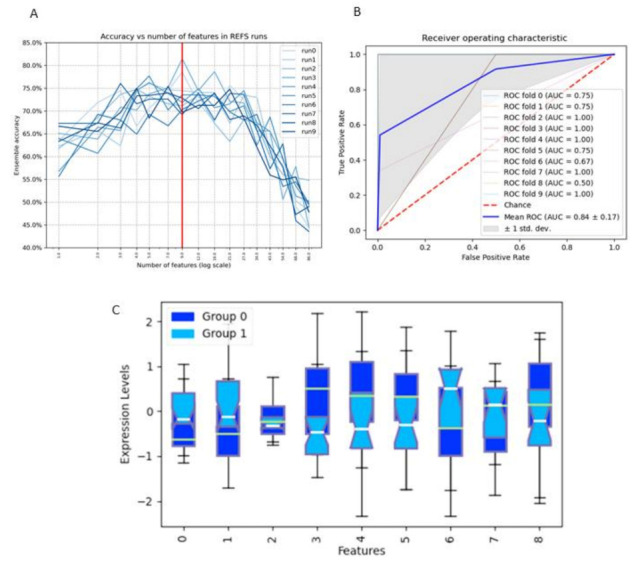
The datasheet of the entire study was analysed regarding marble burying (number of marbles buried) at 8 weeks of age (early adulthood). (**A**) After running the REFS algorithm 10 times, the best signature is at 9 features, with an average global accuracy of all classifiers of 0.73. (**B**) The optimal associated receiver-operating characteristic (ROC) curve for the best-performing classifier Ridge demonstrates an area under the curve (AUC) of 0.84 ± 0.17. (**C**) The 9 features separating the two labels: 0 = Adlercreutzia, 1 = DH: tryptophan (nmol/gr), 2 = Lachnospiraceae Other, 3 = Dehalobacterium, 4 = Th17 cells in MLN, 5 = Activated Th2 cells in spleen, 6 = α-diversity (Shannon index), 7 = PFC: noradrenaline (nmol/gr), 8 = PFC: noradrenaline (nmol/gr), and 9 = Th1 cells in MLN. Label 0: number of buried marbles <10. Label 1: number of buried marbles ≥10.

**Figure 11 nutrients-14-00173-f011:**
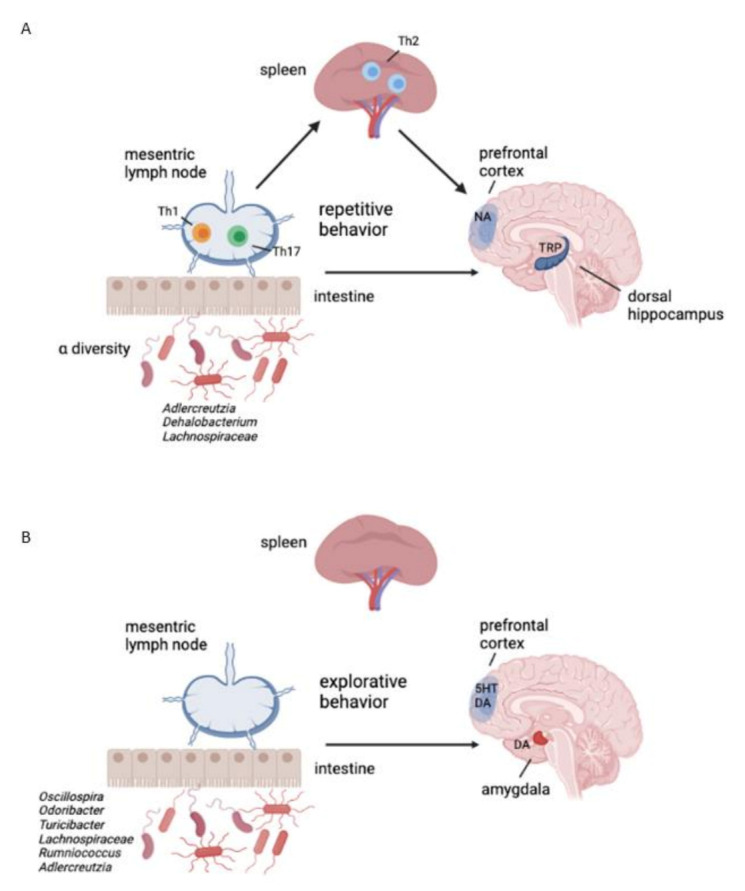
Indications of which feature combination significantly influence repetitive and anxiety-like behaviour and explorative behaviour evaluated by marble burying (**A**) and open field behaviour (**B**) tests, respectively. (**A**) The alpha-diversity and relative abundances of the genera Adlercreutzia and Dehalobacterium, changes in Th1 and Th17 cells in MLN, changes in activated Th2 cells in spleen, and tryptophan levels in dorsal hippocampus significantly predict changes in repetitive behaviour. (**B**) The relative abundances of the genera Oscillospira, Ruminococcus, Odoribacter, Turibacter, Lachnospiraceae other/Unclassified and Adlercreutzia and changes in serotonin and dopamine metabolism in PFC and amygdala significantly predict changes in explorative behaviour. DA: dopamine; 5HT: serotonin; NA: noradrenaline; Th: Thelper; TRP: tryptophan.

**Figure 12 nutrients-14-00173-f012:**
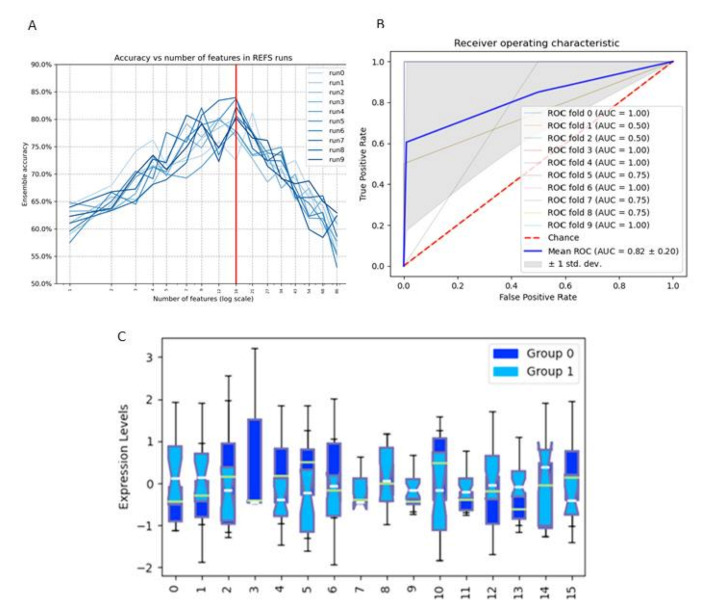
The datasheet of the entire study was analysed regarding the open field test (number of entries in the open field) at 8 weeks of age (early adulthood). (**A**): After running the REFS algorithm 10 times, the best signature is at 16 features with average accuracy of all classifiers of 0.77. (**B**): The optimal associated ROC curve for the best-performing classifier Ridge demonstrates an area under the curve (AUC) of 0.82 ± 0.20. (**C**): The 16 features separating the two labels: 0 = Cyanobacteria Unclassified, 1 = Oscillospira, 2 = Odoribacter, 3 = Turicibacter, 4 = AM: (DOPAC + HVA)/DA, 5 = PFC: 5HIAA/5HT, 6 = Lachnospiraceae Unspecified, 7 = Allobaculum, 8 = DH: noradrenaline (nmol/gr), 9 = Lactobacillus, 10 = PFC: 5-HIAA (nmol/gr), 11 = Lachnospiraceae Other, 12 = Ruminococcus, 13 = Adlercreutzia, 14 = PFC: HVA (nmol/gr), and 15 = AM: (DOPAC + HVA + 3MT)/DA. Label 0: number of entries to open field ≥ 10. Label 1: number of entries to open field < 10.

**Table 1 nutrients-14-00173-t001:** Group analysis of similarity (ANOSIM) results for mouse cecum content microbiota compositions.

Comparison—Genus Level	*n*	Global *R*	*p*-Value ^a^
Control vs. scGOS:lcFOS	8	0.600	0.001
Control vs. n-PUFA	10	0.518	0.001
Control vs. Combination	10	0.817	0.001
scGOS:lcFOS vs. Combination	10	0.366	0.002
n-PUFA vs. Combination	10	0.573	0.001

^a^ = *p* < 0.05; global *R* comparison was based on ANOSIM performed within the software package Primer7; *p*-values were calculated based on a permutational analysis, employing 999 permutations; square root transformation.

**Table 2 nutrients-14-00173-t002:** Feature reduction in marble burying test (repetitive/anxiety-like behaviour).

Feature	Label 1 vs. Label 0
*Coriobacteriaceae Adlercreutzia*	
DH: tryptophan (nmol/gr)	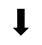
*Lachnospiraceae* Other	-
*Dehalobacteriaceae Dehalobacterium*	
Th17 cells in MLN	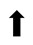
Activated Th2 cells in spleen	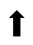
α-diversity (Shannon index)	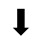
PFC: noradrenaline (nmol/gr)	-
Th1 cells in MLN	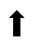

Label 0: number of buried marbles <10. Label 1: number of buried marbles ≥10. Arrow indicates if the feature is reduced or enhanced when the number of buried marbles is reduced. DH: dorsal hippocampus; MLN: mesenteric lymph nodes; PFC: prefrontal cortex; Th: Thelper: ↑ increase; ↓: decrease.

**Table 3 nutrients-14-00173-t003:** Feature reduction open field test (explorative behaviour).

Feature	Label 0 vs. Label 1
Cyanobacteria (c_4C0d; o_YS2)	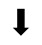
*Ruminococcaceae Oscillospira*	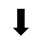
*Odoribacteraceae Odoribacter*	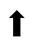
*Turicibacteraceae Turicibacter*	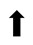
AM: (DOPAC + HVA)/DA	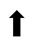
PFC: 5HIAA/5HT	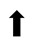
*Lachnospiraceae* Unspecified	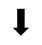
*Erysipelotrichaceae Allobaculum*	-
DH: noradrenaline (nmol/gr)	-
*Lactobacillaceae Lactobacillus*	-
PFC: 5-HIAA (nmol/gr)	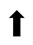
*Lachnospiraceae* Other	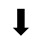
*Ruminococcaceae Ruminococcus*	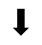
*Coriobacteriaceae Adlercreutzia*	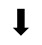
PFC: HVA (nmol/gr)	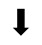
AM: (DOPAC + HVA + 3MT)/DA	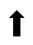

Label 0: number of entries to open field ≥10. Label 1: number of entries to open field <10. Arrow indicates if the feature is reduced or enhanced when the number of entries to open field is enhanced. AM: amygdala; DA: dopamine; DH: dorsal hippocampus; DOPAC: 3,4-dihydroxyphenyl-acetic acid; 5-HIAA: 5-hydroxyindoleacetic acid; 5HT: serotonin; HVA: homovanillic acid; MLN: mesenteric lymph nodes; 3-MT: 3-Methoxytyramine; PFC: prefrontal cortex; ↑: increase; ↓: decrease.

## Data Availability

Microbiota raw sequence data (FASTQ files) were deposited in the National Center for Biotechnology Information (NCBI) Sequence Read Archive (SRA), under the BioProject identifier PRJNA701436.

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
