# Peer review of "Dietary Supplementation throughout Life with Non-Digestible Oligosaccharides and/or n-3 Poly-Unsaturated Fatty Acids in Healthy Mice Modulates the Gut–Immune System–Brain Axis"

_nutrients, 2021, doi:10.3390/nu14010173_

Round 1

Reviewer 1 Report

The study by Szklany and coworkers investigates the influence of tuna oil (n3-PUFA) on the cecal microbiota diversity and on various physiological and behavioural traits during postnatal mouse development.  Interestingly, this study reveals an interaction between nutrition, microbiota diversity and behavioural traits (social and anxiety-like bahavior), which might be linked to the microbiota’s influence on tryptophan metabolism (enterochromaffin cells). Therefore, this study is interesting and clearly within the scope of Nutrients. However, I have a number of comments that need to be addressed.

Comments:

  1. The introduction could be improved. The authors stress the development of the immune system but there are additional morphological traits that are influenced by colonization with gut microbiota (see: Bayer F, Nutrients, 2021). These changes should briefly be mentioned in the introduction. Clearly, the description on immune maturation could be shortened and the description on the involvement of the microbiota in behavioural traits should be extended. Moreover, since the authors put focus on enterochromaffin cells and the microbiota’s influence on tryptophan metabolites, an overview on the indole, serotonin and kynurenine pathway is required and the microbiota’s impact should be described.

  1. What was the rationale of comparing the 3 diets? Please explain clearly in the abstract. Please also provide an overview table on the nutrient contents of each diet.

  1. How do the authors explain a reduced F/B ratio in n-3 PUFA diet, which is increased at Western diet conditions (Kiouptsi K, mBio, 2019) and obesity (Ley RE, Proc Natl Acad Sci USA)? Did other n-3-fatty acid feeding studies also see a shift towards a reduced F/B ratio in cecal content (see: Todorov H, Nutrients, 2020). Please discuss.

  1. In contrast to enterochromaffin cells of the secretory lineage, how did goblet cell counts and Paneth cell counts behave at the different dietary conditions in the jejunum, ileum, and colon? This analysis is necessary to get a complete picture.

  1. Please provide a scheme on the gating strategy for the flow cytometry analyses presented in Figure 7. Also provide representative histograms for these analyses.

  1. How were MLNs isolated? Please describe the protocols more precisely.

  1. The anxiety-like behaviour test figure is mislabelled. Presumably it is not Figure 2 but Figure 9. Also avoid a results description in the legend but rather describe exactly what was analyzed.

  1. The discussion is very narrative and lengthy. I recommend to re-write this part to become more focussed and concise. The role of microbiota-related tryptophan metabolites on social and anxiety-like mouse behaviour should be discussed in the light of existing literature.

  1. I suggest to avoid terms like “super healthy host”.

Author Response

We thanks the reviewer for the useful comments on our manuscript “Dietary supplementation throughout life with prebiotics and/or n-3 poly-unsaturated fatty acids in healthy mice modulates gut-immune system-brain axis”. We are pleased with the reviewers' recognition of our interesting results.
Please, find in the pdf file our point-by point responses to the reviewers’ comments.

Reviewer 2 Report

The manuscript of Szklany and colleagues is a well described work. I only have some suggestions about the n-3 PUFAs supplementation where it seems that the authors are not very familiar.

For example, no data was collected regarding the PUFAs metabolites, but only related to SCFAs. Similarly, PUFAs integration in brain or in blood tissues was not analyzed. These data could have helped explain the results, so I suggest that this be included as a limitation of the study. And for this reason the sentence at lines 679-681 is not supported by the data, so I recommend removing it.

Moreover, n-3 PUFAs are candidate prebiotics by ISAPP, so I suggest to call the two dietary interventions with the proper name to avoid confusion. 

Finally, a more detailed contextualization of the results for PUFAs in the light of the recent papers is needed.

Author Response

We thank the reviewer for the useful comments on our manuscript “Dietary supplementation throughout life with prebiotics and/or n-3 poly-unsaturated fatty acids in healthy mice modulates gut-immune system-brain axis”. We thank the reviewer for the acknowledgement of the well described work.

Please, find in the pdf file our point-by-point responses to the reviewers’ comments. 

Reviewer 3 Report

Authors aimed to determine if prebiotics and omega-3 poly-unsaturated fatty acids (n-3 PUFAs) affect immune function, brain development and behavior in mice. They treated BALB/cByJ mice with a dietary combination of scGOS:lcFOS and n-3 PUFA. Authors did not find differences neither in the immune system or behavior, but found differences in the caecal content microbial community between treatments.

There are several comments that authors may address.

There is a lack of details in the description of treatments. What was the frequency of the treatments? How mice were treated after weaning? How pups were treated before weaning? It seems that pups were treated via their mother, but this is not mentioned in the text. Was the mother treated by gavage? When feces were collected?

Add a brief description of marble burying test and self-grooming and behavior tests.

Figure 4. What type of samples were analyzed in the Figure 4? What of these groups are acetate-, butyrate- and propionate-producing bacteria? This could help to better understand data from the Figure 5.

Figure 5. Is there a correlation between the SCFA-producing bacteria and the type of treatment in the gut and feces?

The rationale to assess the effect of the diets on the 5-HT cells is missing.

Figure 6. There are differences in the effects of diets depending on the part of intestines. Discuss this finding. It is not clear from what diet come the images from the panels D, E, and F.

The connection between 5-HT cells and T-cells is not clear. Also, the rationale of determining the effects of diets on monoamines is not intuitive for a non-expert in this field.

Figure 9 is labeled as Figure 2.

Discussion.

“The present study demonstrates that dietary supplementation with the combination of scGOS:lcFOS and n-3 PUFAs leads to […] a balanced immune response compared to the individual food components.” Is this statement supported by data?

“In our previous work (22), the scGOS:lcFOS effects on the in intestinal microbiota were associated with improved social behaviour and reduced anxiety-like and stereotypic behaviour assessed by marble burying and self-grooming behavioural tests. In this study, the effects of scGOS:lcFOS on these behaviours are less clear.” Why authors think that these differences occur between both studies?

Author Response

We thank the reviewer for careful read through the manuscript and for the comments.

Please, find in pdf file our point-by-point reply responses to the reviewers’ comments.

Round 2

Reviewer 1 Report

The authors have adequately addressed my previous comments. 

Reviewer 3 Report

Authors replied all the comments.